

# ESD Reviews: Evidence of multiple inconsistencies between representations of terrestrial and marine ecosystems in Earth System Models

Félix Pellerin[1], Philipp Porada[2], and Inga Hense[1]

[1]Institute of Marine Ecosystem and Fishery Science (IMF), Center of Earth System Research and Sustainability (CEN), University of Hamburg, Große Elbstraße 133, 22767 Hamburg, Germany
[2]Institute of Plant Science and Microbiology, Center of Earth System Research and Sustainability (CEN), University of Hamburg, Ohnhorststraße 18, 22609 Hamburg, Germay

**Correspondence:** Félix Pellerin (felix.pellerin@uni-hamburg.de)

**Abstract.** Terrestrial and marine ecosystems interact with other Earth system components through different biosphere-climate feedbacks that are very similar among ecosystem types. Despite these similarities, terrestrial and marine systems are often treated relatively separately in Earth System Models (ESM). In these ESM, the ecosystems are represented by a set of biological processes that are able to influence the climate system by affecting the chemical and physical properties of the environment.

While most of the climate-relevant processes are shared between ecosystem types, model representations of terrestrial and marine ecosystems often differ. This raises the question whether inconsistencies between terrestrial and marine ecosystem models exist and potentially skew our perception of the relative influence of each ecosystem on climate. Here we compared the terrestrial and marine modules of 17 Earth System Models in order to identify inconsistencies between the two ecosystem types. We sorted out the biological processes included in ESM regarding their influence on climate into three types of biosphere-

climate feedbacks (i.e. the biogeochemical pumps, the biogeophysical mechanisms and the gas and particle shuttles), and critically compare their representation in the different ecosystem modules. Overall, we found multiple evidences of unjustified differences in process representations between terrestrial and marine ecosystem models within ESM. These inconsistencies may lead to wrong predictions about the role of biosphere in the climate system. We believe that the present comparison can be used by the Earth system modeling community to increase consistency between ecosystem models. We further call for the

development of a common framework allowing the uniform representation of climate-relevant processes in ecosystem modules of ESM.

## 1 Introduction

Terrestrial and marine ecosystems have been mostly independently studied (Steele, 1991; Raffaelli et al., 2005; Menge et al.,

2009). The separate development of terrestrial and marine ecosystem models led to important differences in their conceptual-





ization and complexity. For instance, the vegetation models consider $CO_2$ as one of the main limiting factors for plant growth and therefore include a detailed representation of photosynthesis (reviewed in Rogers et al., 2017), while nutrient limitation is still ignored or simplistic in most of the models (Fisher et al., 2014). Conversely, the marine ecosystem models estimate growth of primary producers based on nutrient and light limitations, and ignore the processes behind carbon uptake (reviewed

in Laufkotter et al., 2015). These differences can originate from intrinsic differences between terrestrial and marine habitats (e.g. physical environment gaseous vs liquid), but can also be unjustified (e.g. nutrient limitation, Elser et al., 2007). Since the development of Earth System Models (ESM) in which both terrestrial and marine models are combined, it is not clear whether inconsistencies (i.e. unjustified differences) between terrestrial and marine ecosystem models exist and potentially skew our perception of the relative influence of each ecosystem on climate regulations. For instance, the absence of nutrient limitation in

terrestrial models may lead to qualitatively different responses of marine and terrestrial ecosystems to elevated $CO_2$, although the response may be similar in reality.

In ESM, marine and terrestrial ecosystems are mechanistically represented by different biological processes that can both respond to and influence climate. These climate-biosphere feedbacks involve biogeochemical and biogeophysical mechanisms (e.g. Hungate and Hampton, 2012; Hense et al., 2017; Bonan and Doney, 2018) that are, in many cases, similar for terres-

trial and marine ecosystems. Terrestrial and marine ecosystems are both involved in biogeochemical cycles (e.g. carbon cycle, Heimann and Reichstein (2008); Bloom et al. (2016); Worden et al. (2015); nitrogen cycle, McNeill and Unkovich (2007); Gruber and Galloway (2008); Canfield et al. (2010); Zehr and Kudela (2011); phosphorous cycle, Filippelli (2002)) and bio-geophysical feedbacks (e.g. albedo/light absorption, Dickinson and Hanson (1979); Frouin and Iacobellis (2002); Park et al. (2015); Alkama and Cescatti (2016); Zeng et al. (2017); momentum, Jöhnk et al. (2008); Sonntag and Hense (2011); Alkama

and Cescatti (2016); Zeng et al. (2017); aerosol emissions, Andreae and Crutzen (1997); Meir et al. (2006)). However, the representation of these mechanisms differs between terrestrial and marine modules of Earth System Models (see Fisher et al. (2014) for a review of the terrestrial ecosystem models and Hense et al. (2017) for a review of the marine modules of ESM). Improving consistency in the representation of the biosphere-climate feedbacks between the two types of ecosystem in ESM is necessary to understand the central role that the biosphere plays in the earth system.

Although the biosphere in ESM is constituted by the combination of terrestrial and marine ecosystems, previous comparative studies were focusing on one or the other ecosystem model individually (e.g. Schwalm et al., 2010; Huntzinger et al., 2012; Cabré et al., 2015; Laufkotter et al., 2015; Fisher et al., 2014). To our knowledge, there is no study that critically evaluates whether marine and terrestrial models represent the processes occurring in both ecosystem types in a consistent way. Therefore, the present study aims to compare, for the first time, the terrestrial and marine modules of Earth System Models in order to

identify inconsistencies between the two ecosystem types. Inconsistencies refer to unjustified differences in the representation of particular processes between ecosystem models. They cover: 1) the representation of a process in one model and its absence of representation in the second model, 2) different levels of complexity in the representation of a given process between models and 3) different modeling approaches used to represent the same process. Note that most of the inconsistencies identified in the present study correspond to the first category (i.e. presence of a process in one ecosystem model and not in the other).





The absence of important climate relevant processes in terrestrial (or marine) modules can offset the benefit of an accurate representation of the corresponding processes in marine (or terrestrial) ecosystem modules.

We review and critically compare the terrestrial and marine modules of 17 Earth System Models that have been applied to the CMIP5 and CMIP6 experiments (Table 1, Taylor et al., 2012; Eyring et al., 2016). These 17 ESM included 15 and 16 distinct terrestrial and marine ecosystem models, respectively (Table 1). The different climate-relevant processes included in
these ESM were listed and sorted regarding their function and their influence on climate (Table 2).

## 2   The biosphere – climate feedbacks

We present here a detailed description of the main biosphere-climate feedbacks, thereby differentiating the biogeochemical and the biogeophysical influence of biosphere on climate. The term climate refers to multiple aspects of the climatic system, such as average temperature, humidity and seasonality. Here we focus mainly on the effect of biosphere on surface and atmospheric
temperature. We provide a detailed comparison of how biosphere-climate feedbacks are represented in terrestrial and marine ecosystem modules of current ESM in the next section.

Hense et al. (2017) described the biogeochemical and biogeophysical influence of the biosphere on climate for marine ecosystems by three main mechanisms: the biogeochemical pumps, the biogeophysical mechanisms and the gas and particles shuttles. In the following we also apply these three mechanisms to terrestrial systems.
The biogeochemical pumps influence the cycles of multiple chemical elements, including carbon, through the biosphere. The growth of organisms depends on chemical elements and ions that are thus removed from the environment and stored in the organic matter before being released during exudation, excretion, respiration and remineralization of dead organic matter (Luo and Weng, 2011; Worden et al., 2015). The processes involved in the uptake, storage and release of these elements are thus important for the biogeochemical cycles and affect the climate by modifying the concentration of these elements in the
environment, such as carbon in the atmosphere (Heimann and Reichstein, 2008; Monroe et al., 2018). Both terrestrial and marine ecosystems are involved in the biogeochemical pumps. For instance, terrestrial ecosystems were responsible for the net uptake of 3.2 giga tons of carbon (GtC) per year over the period 2009-2018 (Friedlingstein et al., 2019). In the same study they estimated that the ocean contributed to the uptake of 2,5 GtC per year, with approximately 90% of this uptake due to the biological pump (2.25 GtC per year) and the remaining by the solubility pump (Sarmiento and Gruber, 2006; Boyd et al.,
2019). This corresponds to 29 and 20% of the human $CO_2$ emissions for terrestrial and marine ecosystems respectively. Besides carbon, terrestrial and marine ecosystems are also involved in the cycle of nitrogen (McNeill and Unkovich, 2007; Gruber and Galloway, 2008; Canfield et al., 2010; Zehr and Kudela, 2011), phosphorus (Filippelli, 2002), iron (Boyd and Ellwood, 2010; Hutchins and Boyd, 2016; Tagliabue et al., 2017; Wu et al., 2019), sulfur (Turner et al., 2016; Wasmund et al., 2017), silicon (Struyf et al., 2009) and oxygen (Walker, 1980). These elements can directly affect climate or be coupled to carbon
via stoichiometric ratios and affect the climate indirectly (Elser et al., 2010; Meunier et al., 2017; Zhu et al., 2020). Terrestrial ecosystems play also a crucial role on the water cycle and marine ecosystems influence the carbonate cycle (Ridgwell and Zeebe, 2005).





The biogeophysical mechanisms include the influence of the biosphere on its surrounding physical environment. Organisms modify the optical, mechanical and thermal properties of the environment, both on land and ocean. On land, vegetation influences surface albedo, roughness and evapotranspiration, and, as a consequence, wind speed, surface temperature and humidity (e.g. Hollinger et al., 2010; THOM, 1971; Fisher et al., 2014; Zeng et al., 2017). The different biogeophysical impacts of vegetation can influence climate in opposite directions. For instance, the recent increase in leaf area index (LAI) following the rise of atmospheric $CO_2$ led to a decrease in albedo, causing higher surface temperature (Betts, 2000). Conversely, the increase in LAI enhances transpiration which cools the surface via the latent heat flux (Shen et al., 2015). Altogether, the biogeophysical mechanisms have buffered climate change globally during the past 30 years (Zeng et al., 2017). In oceans, phytoplankton communities are also influencing surface albedo, temperature distribution, air-sea gas exchange, turbulent viscosity and vertical mixing (Frew et al., 1990; Sathyendranath et al., 1991; Sonntag and Hense, 2011; Katija and Dabiri, 2009). Indeed, due to surface distribution, phytoplankton, especially reflective species (e.g. coccolithophores) and positively buoyant species (e.g. filamentous cyanobacteria), influence surface albedo (Kahru et al., 1993; Tyrrell et al., 1999; Frouin and Iacobellis, 2002; Jung and Moon, 2019) and also trap the heat at the surface of the ocean, impacting sea surface temperature and ocean circulation globally (e.g. Paulsen et al., 2018).

Finally, the biosphere emits gas and particles into the atmosphere and thus influences the concentrations of green house gases and aerosols, impacting radiative forcing, cloud formation and light transmission through the atmosphere. Some processes involved in the gas and particle shuttle mechanisms are also influencing the biogeochemical pumps and the biogeophysical mechanisms. However, we distinguish this biosphere-climate feedback from the other to account for the direct influence of biosphere on atmospheric composition and properties. Marine and terrestrial ecosystems emit greenhouse gases, including $CO_2$, water vapor, methane and nitrous oxide, into the atmosphere (Meir et al., 2006; Rap et al., 2013; Stocker et al., 2013). Photosynthesis, autotrophic and heterotrophic respiration drive the exchanges of $CO_2$ between the atmosphere and biosphere (e.g. Heimann and Reichstein, 2008). On land, the exchange of water from plant to atmosphere is also essential for the water cycle and water vapor concentration in the atmosphere (e.g. Davin and de Noblet-Ducoudre, 2010). The emission of methane ($CH_4$) by the biosphere occur mainly in anoxic region of the soil, typically in wetland, during the decomposition of organic matter (see Dean et al., 2018, for a review of the different biologic methane sources and sinks), but also, to a lesser extent, in the ocean (Valentine, 2011). Finally, nitrous oxide ($N_2O$) is produced during nitrification and denitrification processes by bacteria in soil and ocean. $N_2O$ is a greenhouse gas at least 200 times more potent than $CO_2$ (Gruber and Galloway, 2008; Myhre et al., 2014; Battaglia and Joos, 2018), and is also involved in the depletion of ozone (Myhre et al., 2014).

Besides greenhouse gases, significant quantities of aerosols are emitted by the biosphere. Aerosols affect climate as they influence cloud formation and properties, and also scatter the incoming beam solar radiation (Novakov and Penner, 1993; Andreae and Crutzen, 1997). The scattering of solar radiation could further affect climate by promoting plant growth which benefit from diffuse radiations (Rap et al., 2018). Biosphere-emitted aerosols, called biogenic volatile organic compounds (BVOC), include a variety of chemical substances as well as cells and organic fragments emitted by both kind of ecosystems Meir et al. (2006). One of the main BVOC produced by marine ecosystems is the dimethyl sulphide (DMS). DMS emission into the atmosphere may promote cloud formation, increasing albedo and cooling down the surface temperature (e.g. Charlson et al.





(1987) but see Quinn and Bates (2011)). On land, the emission of BVOCs may lead to more complex climate responses. Indeed, some BVOC may promote ozone formation and thus increase greenhouse effect, while others may form cloud condensation

nuclei and increase albedo (Peñuelas and Staudt, 2010).

The three mechanisms described above are able to affect climate in different directions by either buffering or accentuating changes in the climatic conditions. For instance, contemporary climate change affects biological processes which can reciprocally accelerate (i.e positive feedback) or dampen (i.e. negative feedback) climate change. A good example of a negative feedback mechanism is enhanced vegetation carbon uptake following the increase in atmospheric $CO_2$ concentration that may

buffer human induced $CO_2$ emissions and thus surface temperature (Canadell and Raupach, 2008). On the other hand, the increase in temperature is expected to enhance soil bacterial metabolism and may therefore increase natural $CO_2$ and methane production by micro-organisms (Montzka et al., 2011; Dean et al., 2018), increasing further more atmospheric temperature (i.e. positive feedback). These two examples highlight the central role of the biosphere on climate regulation and the necessity to accurately represent biosphere-climate feedbacks on both land and ocean to predict future (and past) climate change.

## 2.1   Comparative review of terrestrial and marine ecosystem models in ESM

To compare terrestrial and marine ecosystem models, we listed all the climate-relevant processes that are involved in the biosphere-climate feedbacks described in the previous section. We focused on the processes that were present at least in one of the 15 terrestrial ecosystem models and 16 marine ecosystem models we reviewed. Based on this list, we compared how the different biosphere-climate feedbacks are considered in terrestrial and marine ecosystem modules of current ESM (Table 2).

Overall, inconsistencies exist among terrestrial and marine ecosystem models of ESM regarding the biogeochemical, the biogeophysical and, to a lesser extent, the gas and particle shuttle mechanisms (Table 2, Fig. 2). In the next paragraphs, we describe these inconsistencies among ecosystem modules in ESM by providing a detailed comparison of each process, as well as its potential influence on - and its response to - climate.

### 2.1.1   Biogeochemical processes

The biogeochemical pump, including the carbon cycle, is the mechanism represented with the most diverse processes in terrestrial and marine modules of Earth System Models. However, terrestrial and marine models show important differences (Table 2, Fig. 2), due to the way they were initially developed. Because terrestrial ecosystem models were initially based on the carbon cycle, the processes of photosynthesis, respiration, phenology, mortality and soil respiration (i.e. C remineralization in Table 2) are represented in the 15 terrestrial modules. The net primary production is calculated in each model from the balance

of photosynthesis and respiration, each of these processes being represented by a set of physiological traits interacting with the environment (Fig. 1).

Besides carbon, the representation of other important elements such as nitrogen, phosphorus and iron in terrestrial ecosystem models is rather sparse compared to marine models, while the different ecosystem types are similar in terms of N and P limitation (Elser et al., 2007). Nitrogen is an essential element of protein, including the photosynthetic enzymes such as Rubisco,

the enzyme that fixes carbon from the atmosphere into carbohydrates. Phosphorus is also important for plant physiology as it





is part of the composition of nucleid acid, lipid and bioenergetic molecules such as ATP (Wright et al., 2004). For that reason, the terrestrial modules used for the CMIP6 experiments include nitrogen limited carboxylation capacity of the plants. The last generation of land surface models incorporate a representation of the phosphorous cycle as well (Fisher and Koven, 2020). However, these land surface models are not integrated in ESM and the representation of the phosphorus cycle is still needed in

most of the terrestrial ecosystem models within ESM.

In contrast, the development of marine ecosystem models was initially based on the cycles of limiting nutrients for phyto-plankton growth. In the simplest models, only one element was explicitly modeled and the concentrations of others such as carbon were calculated using a fixed ratio (i.e. the Redfield ratio, Redfield, 1934). However, the ratios of the main nutrients and elements (N, C, P, Si, Fe) vary among phytoplankton groups and with the environmental conditions (Rhee.G.Yull, 1978;

Goldman et al., 1979; Geider and La Roche, 2002; Quigg et al., 2003). Therefore, the use of a fixed ratio among elements constrains the predictions within a narrow range of potential ecosystem responses that may not represent the current states of marine ecosystems. Some recent marine ecosystem models (e.g. BFMv5.2, Vichi et al., 2015) thus included variable ratios that are influenced by the environmental conditions (i.e. the availability of the different elements in the environment). Nevertheless, the explicit carbon cycle representation is still lacking in most of the models and the processes of photosynthesis and cell

respiration are ignored (Table 2, Fig. 1).

Another difference between terrestrial and marine models in Earth System Models is the representation of trophic interac-tions (Table 2, Fig. 2). In terrestrial models, only plants and soil microorganisms are represented while higher trophic levels are ignored. Although it has been traditionally assumed that trophic interactions had a limited effect on large scale climate, recent studies underline the potential strong influence of grazers and higher trophic levels on ecosystem carbon uptake and storage

(e.g. Schmitz et al., 2018). For instance, the disturbance induced by elephants enhance aboveground biomass, and thus carbon storage, in African tropical forest (Berzaghi et al., 2019). Grazers may also influence climate by changing the biogeophysical properties of the ecosystem. For instance, larger animals grazing on boreal ecosystem limit shrub height and density, cooling down air temperature by increasing summer albedo (Te Beest et al., 2016), or protect the permafrost from thawing (Beer et al., 2020).

Most of the marine models reviewed here considered zooplankton, heterotrophic organisms that feed on primary producers. Zooplankton has been implemented in marine ecosystem models because they exert a strong grazing pressure on phytoplankton, impeding or buffering phytoplankton bloom formation (Prowe et al., 2012). Zooplankton and higher trophic levels in general, play also an important role in carbon removal (Davison et al., 2013; Steinberg and Landry, 2017), nutrient distribution and recycling (Vanni, 2002; Schmitz et al., 2010). However, the representation of the trophic chains in marine models remains very

simple. Complex trophic interactions can strongly influence carbon storage by marine ecosystems. For instance, Wilmers et al. (2012) showed that the presence of sea otters increases the carbon fixation by kelp by a factor of 10 due to their predation pressure on kelp grazers. A better understanding of the importance of trophic complexity and length on carbon cycle and on climate is needed to properly judge the necessity to include them in Earth System Models.

There are also several justified differences between terrestrial and marine ecosystem which are represented in ESM (Table 2).

Terrestrial and marine ecosystem models differ in the representation of processes involved in the water cycle and silicate cycle.





The role of the biosphere on the carbon cycle is central and has been considered in terrestrial and marine models of ESM. Both ecosystem models are based on the growth of photosynthetic primary producers (i.e. net primary production, NPP). However, important differences exist in the representation of growth and its implication for the carbon cycle among ecosystem types.

In terrestrial ecosystem models, NPP corresponds to the difference between the carbon fixed by photosynthesis and the carbon released by respiration. Photosynthesis has been modeled using three different approaches in terrestrial ecosystem models (Arora, 2002; Fisher et al., 2014): the biochemical approach, the light use efficiency approach and the carbon assimilation approach. Most of the modules reviewed here followed the biochemical approach to represent photosynthesis (a). In this approach, the rate of $CO_2$ assimilation is limited by i) the rate of carboxylation by Rubisco, ii) electron transport which depends on light and iii) the transport of photosynthetic products (Farquhar et al., 1980; Collatz et al., 1992). Note that the transport of photosynthetic products is sometimes ignored in the present modules (a). In the biochemical approach, the assimilation of carbon by photosynthesis is closely linked to stomatal conductance that control intercellular $CO_2$ concentration and water exchange with the atmosphere. A part of the carbon fixed by photosynthesis is re-emitted to the atmosphere through respiration. In all the models, respiration is divided into maintenance respiration and growth respiration. The growth respiration is a fixed proportion of the NPP, while the maintenance respiration, considered at the organ level (i.e. leaf, stem, root), can depend on temperature, nitrogen content and the rate of carboxylation (a).

In marine models (b), the growth of primary producers ignore the processes of photosynthesis and respiration. NPP depends on light, nutrients and temperature (b). Most of the modules do not explicitly consider carbon, and thus deduce carbon assimilation by applying a stochiometric ratio. For that reason, respiration cannot be properly estimated because the ratio of elements in phytoplankton is considered constant. Only one model that explicitly represents the carbon cycle accounts for respiration by phytoplankton (BFMv5.2).

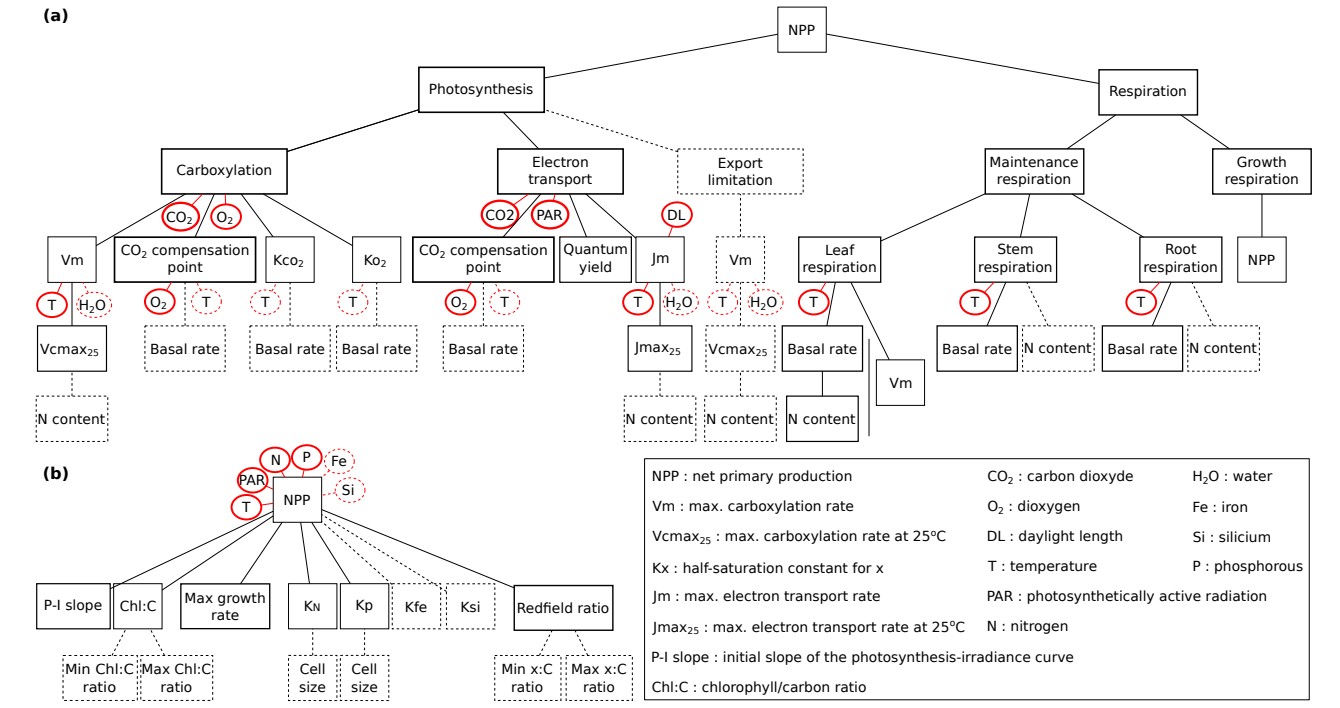

**Figure 1.** Box 1: Differences in the representation of primary producers' carbon growth between terrestrial and marine ecosystem modules of ESM. The text present the main approaches used to model carbon assimilation in terrestrial and marine ecosystem modules. The diagrams detail the different biological traits and processes (boxes) considered in the calculation of primary producers' growth in most of the terrestrial (a) and marine (b) ecosystem modules of ESM. From the bottom to the top, the black lines indicate the influence of different traits on the trait/process above. The influence of the environmental conditions are indicated in red. The traits and environmental factors considered in only few modules are represented by dotted lines and boxes.





These differences rely on biological singularity of terrestrial and marine ecosystems. Water is an essential and limiting element for terrestrial plant growth and is tightly linked to carbon uptake. Similarly, silicon is an important constituent of the shell of an abundant phytoplankton group (i.e. the diatoms). Diatoms are often considered in marine ecosystem models because their silicate shell influence particle sinking from the surface to the deep ocean, and thus influence carbon storage in the ocean.

For these reasons, these differences among ecosystem models are not part of the inconsistencies we identified in the previous paragraphs.

### 2.1.2 Biogeophysical processes

The biogeophysical influence of the biosphere is unequally considered in terrestrial and marine models of Earth system models (Table 2, Fig. 2). In terrestrial modules, there is a good representation of vegetation albedo, light absorption, evapotranspiration

and roughness length. For instance, the calculation of vegetation albedo often considered multiple leaf traits such as absorbance, transmittance, surface and orientation (e.g. Kowalczyk et al., 2013). Global changes, including deforestation, could have a strong impact on the biogeophysical characteristics of land ecosystems (Davin and de Noblet-Ducoudre, 2010). The effect of the biogeophysical changes associated with deforestation may counteract the effects of biogeochemical changes (i.e. lower $CO_2$ fixation by vegetation), at least in certain regions (mostly at high latitude) (Claussen et al., 2001).

In contrast to terrestrial models, the representation of the biogeophysical mechanisms in marine modules of ESM is scarce (Table 2, Fig. 2). Only the light absorption by phytoplankton is considered in 6 of the 16 modules reviewed here, modifying the heat distribution in the water column. All other biophysical mechanisms are ignored. Models including light absorption by phytoplankton obtained contrasted results. They simulated either an average cooling of the surface ocean (Mignot et al., 2013; Paulsen et al., 2018) due to the increase of intra-annual surface temperature variation with light absorption, while others

predicted an increase of sea surface temperature when biologically induced heating is considered (Lengaigne et al., 2009; Patara et al., 2012). This feedback mechanism may become more important in the future (Hense et al., 2013), because higher temperature could increase surface buoyant phytoplankton (cyanobacteria) abundance and thus enhance light absorption. For instance, climate change in the Arctic might be amplified by 20% due to higher light absorption that results from an increase in plankton density caused by warmer water (Park et al., 2015). Further ESM including light absorption by plankton are therefore

needed to reduce prediction uncertainties and better predict climate change. The influence of other biogeophysical mechanisms in the ocean (i.e. surface albedo, air-sea gas exchange, turbulent viscosity and vertical mixing) are more difficult to evaluate because they are less studied and will thus not be discussed further.

### 2.1.3 Gas and particles emission processes

Terrestrial and marine ecosystem models are rather similar in their representation of gas and particle shuttles. Both ecosystem

modules only sparsely consider the role of the biosphere on greenhouse gas and aerosol atmospheric concentration, except $CO_2$ and water vapor in terrestrial ecosystem models (Table 2, Fig. 2).

    The ecosystem models mainly differ with regard to $CO_2$ emissions. While $CO_2$ emission back to the atmosphere through plant and soil respiration is well considered in terrestrial ecosystem models, the influence of marine biota on dissolved and




**Figure 2.** General scheme summarizing the representation of the different biosphere-climate feedbacks in terrestrial and marine ecosystem modules of current ESM. The different processes are sorted into the biogeochemical pumps (1), the biogeophysical mechanisms (2) and the gas and particle shuttles (3). The ratio of the number of modules in which each process is present on the total number of reviewed module (15 terrestrial distinct terrestrial ecosystem modules and 16 distinct marine ecosystem modules) is indicated for each ecosystem model type.

atmospheric $CO_2$ is poorly represented (Table 2, Fig. 2). The great majority of the models do not explicitly consider the

processes of photosynthesis and respiration (Fig. 1). We now know that the reaction norm of photosynthesis and respiration to





temperature are different and that organisms adapt and both rates change with higher temperature (Padfield et al., 2016; Schaum et al., 2017; Barton et al., 2020). The resulting net exchange of carbon between atmosphere and marine biota is thus sensitive to temperature. By ignoring photosynthesis and respiration processes, marine models may fail to properly predict the influence of phytoplankton on the carbon cycle, and thus on atmospheric $CO_2$ concentration under changing climatic conditions.

$CH_4$ emissions are considered only in 5 of the 15 terrestrial modules and in none of the marine modules (Table 2, Fig. 2). Knowing that the large majority of $CH_4$ is produced by terrestrial ecosystems (Dean2018) and that soil production is predicted to increase under future conditions (Montzka et al., 2011; Dean et al., 2018), $CH_4$ emissions in terrestrial modules of ESM are still underrepresented. Ocean methane production is assumed to be small compared to terrestrial ones (Valentine, 2011), thus the absence of methane emission in marine ecosystem models of current ESM may be justified. Nevertheless, the $CH_4$

emission by coastal ecosystems might also grow under future climatic conditions (Al-Haj and Fulweiler, 2020) and further evaluation of their role in global $CH_4$ emissions will be needed to judge the necessity to include them in ESM.

    $N_2O$ production is represented in a quarter of the terrestrial and marine modules of current ESM (Table 2, Fig. 2). While the estimations of $N_2O$ emissions are still rather uncertain (from 3,3 to 9 Tg N $y^{-1}$ for terrestrial ecosystems and from 1,8 to 9,45 Tg N $y^{-1}$ for marine ecosystems, Ciais et al., 2014), it has been shown that $N_2O$ plays an important role in the past and ongoing

climate change (Schilt et al., 2010; Stocker et al., 2013). Furthermore, climate change is affecting biologically mediated $N_2O$ emissions (Stocker et al., 2013; Martinez-Rey et al., 2014) and also the overall impact of $N_2O$ on climate. Increasing temperature may enhance the transport of $N_2O$ from its source location (Earth surface) to its sink location (stratosphere), reducing both the lifetime of $N_2O$ and its global warming potential (Kracher et al., 2016). ESM represent the adequate tools to study such complex feedbacks between biosphere, atmosphere and climate. However, it necessitates a good representation of the

processes behind $N_2O$ emissions that is currently missing in many current ESM (in both terrestrial and marine modules and despite their consistency regarding this particular process).

    The emission of aerosols is represented in the form of biogenic volatile organic compound (BVOC, 3/15 modules) in terrestrial ecosystems and dimethyl sulphide (DMS, 6/16 modules) in marine ones (Table 2, Fig. 2). The fact that there is consistent evidence that DMS decreases atmospheric temperature (e.g. Charlson et al. (1987); McCoy et al. (2015) but see Quinn and

Bates (2011)) may explain the slightly higher number of marine modules representing volatile production than terrestrial ones in ESM. Future changes of DMS-fluxes may be stronger under ocean acidification as suggested by Six et al. (2013) and Schwinger et al. (2017). However, only one marine ecosystem module considered the pH dependency of DMS production so far (PISCESv2-gas module of CNRM-ESM2.1, Séférian et al., 2019).

    Similarly, BVOC production is sparsely presented in terrestrial modules of ESM (Table 2, Fig. 2). Current knowledge

indicates that the influence of terrestrial aerosols on climate is highly variable and depends on the type of emitted substances (see Peñuelas and Staudt, 2010). Nevertheless, a recent study estimated that aerosol-climate feedbacks could be strong enough to moderate the $CO_2$-related atmospheric temperature increase (Scott et al., 2018). Further studies are needed to identify the most important aerosols emitted by terrestrial ecosystems in order to facilitate their integration in ESM. The future inclusion of BVOC emission in ESM may be further considered knowing that the production of BVOC by vegetation is predicted to

increase under climate change (Laothawornkitkul et al., 2009; Peñuelas and Staudt, 2010; Zhao et al., 2017).



## 3   Discussion and Conclusions

The present review highlights important inconsistencies in the representation of the main biosphere-climate feedbacks between terrestrial and marine ecosystem models within ESM. The main processes related to the carbon cycle (i.e. photosynthesis and respiration) are still implicitly represented in marine models, while nutrient limitation of growth is only scarcely considered in

terrestrial ones. Major differences also occur in the representation of biophysical mechanisms. Light absorption only is partially considered in marine ecosystem models while terrestrial ecosystem models represent adequately the influence of vegetation on land surface biogeophysics (i.e. albedo, roughness length, evapotranspiration and light absorption). Conversely, emission of non-$CO_2$ gases and particles by the biosphere into the atmosphere is consistently represented, even if scarcely, in terrestrial and marine modules of ESM.

Inconsistencies in ecosystem representation can lead to wrong predictions about the role of the biosphere in the climate system. The relative importance of terrestrial and marine ecosystems on climate regulation might be inaccurately perceived because predictions result from different models representing various processes with different complexity levels. As an example, terrestrial ecosystems models currently include a wider range of biosphere-climate feedbacks than marine models (Fig. 2), leading to the potential overweighting of the influence of terrestrial ecosystem on climate compared to marine ones. Future pre-

dictions could be further biased by the lack of mechanistic representation of particular processes in one or the other ecosystem model. While human activities are predicted to bring atmospheric green-house gas concentration above 750ppm $CO_2$ equivalent and temperature above +4°C in the worst-case scenario (RCP8.5, IPCC, 2014), the response of biosphere to these changes and the subsequent feedbacks on climate can be underestimated in marine models due to the lack of explicit representation of photosynthesis and respiration (Fig. 1). Conversely, the predicted increase of $CO_2$ uptake by terrestrial ecosystem following

the increase in atmospheric $CO_2$ and temperature might be overestimated due to the lack of representation of nutrient limitation on plant growth (Zaehle et al., 2015; Wieder et al., 2015; Terrer et al., 2019).

Nevertheless, there is the potential to reduce inconsistencies in the representation of terrestrial and marine ecosystems in ESM in the near future. We believe that more collaboration among terrestrial and marine scientific communities can strongly improve the representation of the biosphere in ESM. By identifying the major inconsistencies that currently exist among

ecosystem modules in ESM, the present work provides a solid basis toward future consistency in biosphere representation. Modelers could further benefit from the development of a common framework allowing to i) identify the important processes involved in the different biosphere-climate feedbacks and ii) standardize their inclusion in both terrestrial and marine ecosystem models of ESM. Such general framework is still missing and thus urgently needed. Finally, the development of combined databases joining both terrestrial and marine organisms could help to remove current inconsistencies in the representation of

these organisms in ESM. The reliability of model predictions depends on the use of accurate parameter values for the different mathematical equations used to represent diverse biological processes. Estimation of parameters at a global scale rely on the development of worldwide large data bases (e.g. the TRY data base for plants, Kattge et al., 2011) which are currently scarce especially for marine ecosystems.





Altogether, we argue that increasing consistency between ecosystem modules of ESM enables to extend their ability to

predict future climate. While human activities strongly impact the biosphere, improving our understanding and representation of the biosphere-climate feedbacks in both terrestrial and marine ecosystem models is crucial to make reliable predictions and build efficient management policies.

*Author contributions.* All authors designed the study. FP reviewed the ESM and wrote the first draft of the manuscript. All authors contributed substantially to revisions.

*Competing interests.* The authors declare that they have no conflict of interest.

*Acknowledgements.* We are grateful to Laurin Steidle, Maike Scheffold and Rémy Asselot for their useful comments on the early version of the manuscript. This study is a contribution to the Cluster of Excellence 'CLICCS - Climate, Climatic Change, and Society', contribution to the Center for Earth System Research and Sustainability (CEN) of Universität Hamburg.



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



**Table 1.** List of the 17 earth system models reviewed in this study. For each ESM, the names of the terrestrial ecosystem module and the marine ecosystem module are indicated as well as the CMIP phase for which the model has been applied and the main references. A total of 15 different terrestrial ecosystem models and 16 marine ecosystem models are considered.

| Earth system model | Terrestrial eco. mod. | Marine eco. mod. | CMIP | References |
|---|---|---|---|---|
| ACCESS-ESM1.5 | CABLE Casa-CNP | WOMBAT | CMIP5 | Law et al. (2017); Kowalczyk et al. (2006); Oke et al. (2013) |
| BNU-ESM | CoLM | iBGC | CMIP5 | Ji et al. (2014); Ji and Dai (2010); Galbraith et al. (2011) |
| CanESM5 | CLASS CTEM | CMOC | CMIP6 | Swart et al. (2019) |
| CESM1 | CLM4.5 | BEC | CMIP5 | Oleson et al. (2013); Moore et al. (2013) |
| CESM2 | CLM5 | MARBL | CMIP6 | Danabasoglu et al. (2020); Lawrence et al. (2019); Moore et al. (2013) |
| CMCC-ESM2 | CLM4.5 | BFMv5.1 | CMIP6 | Cherchi et al. (2019); Oleson et al. (2013); Vichi et al. (2015) |
| CNRM-ESM1 | ISBA | PISCESv2 | CMIP5 | Séférian et al. (2016); Gibelin et al. (2006); Aumont et al. (2015) |
| CNRM-ESM2.1 | ISBA | PISCESv2-gas | CMIP6 | Séférian et al. (2019); Gibelin et al. (2006); Aumont et al. (2015); Martinez-Rey et al. (2014) |
| GFDL-ESM2M/ESM2G | LM3 | TOPAZ2m | CMIP5 | Dunne et al. (2013) |
| HadGEM2 | MOSES - TRIFFID | Diat-HadOCC | CMIP5 | Collins et al. (2011); Cox (2001); Palmer and Totterdell (2001) |
| IPSL-CM5 | ORCHIDEE | PISCESv2 | CMIP6 | Naudts et al. (2015); Aumont et al. (2015) |
| MIROC-ESM | SEIB-DVGM | NPZD | CMIP5 | Watanabe et al. (2011); Sato et al. (2007); Kawamiya et al. (2000) |
| MIROC-ES2L | VISIT-e | OECO-v2 | CMIP6 | Hajima et al. (2019); Ito and Oikawa (2002); Inatomi et al. (2010) |
| MPI-ESM1.2 | JSBACH3.2 | HAMOCC6 | CMIP6 | Mauritsen et al. (2019); Reick et al. (2013); Ilyina et al. (2013) |
| MRI-ESM1 | HAL | unnamed | CMIP5 | Yukimoto et al. (2011) |
| NorESM1 | CLM4 | HAMOCC5 | CMIP5 | Bentsen et al. (2012); Lawrence et al. (2011); Ilyina et al. (2013) |
| UKESM1 | JULES | MEDUSA | CMIP6 | Sellar et al. (2019); Clark et al. (2011); Yool et al. (2013) |

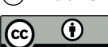



**Table 2.** List of climate-relevant processes and their consideration in the terrestrial (T) and marine (M) modules of the different ESM. The processes are sorted according to their role in the biogeochemical pumps (column 3), the biogeophysical mechanisms (column 4) and the gas and particle shuttles (column 5). The * for N source and sink of the CoLM module indicates that the processes are included but have been turned-off for the CMIP5 experiments. The - indicate that the given process is not relevant for terrestrial or marine ecosystem.

| Module | Eco. | Photosynthesis | Respiration | Phenology | Mortality | C remineralization | Grazing | N source and sink | P source and sink | Fe source and sink | Si source and sink | $H_2O$ source and sink | Albedo | Light absorption | Roughness length | Evapotranspiration | Respiration | Evapotranspiration | $CH_4$ production | $N_2O$ production | Aerosol production |
|---|---|---|---|---|---|---|---|---|---|---|---|---|---|---|---|---|---|---|---|---|---|
| CABLE Casa-CNP | T | x | x | x | x | x |  | x | x |  | - | x | x | x | x | x | x | x |  |  |  |
| CoLM | T | x | x | x | x | x |  | * |  |  | - | x | x | x | x | x | x | x |  |  |  |
| CLASS CTEM | T | x | x | x | x | x |  |  |  |  | - |  | x | x | x | x | x | x | x |  |  |
| CLM4.5 | T | x | x | x | x | x |  | x |  |  | - | x | x | x | x | x | x | x | x | x | x |
| CLM5 | T | x | x | x | x | x |  | x |  |  | - | x | x | x | x | x | x | x | x | x | x |
| ISBA | T | x | x | x | x | x |  |  |  |  | - | x | x | x | x | x | x | x |  |  |  |
| LM3 | T | x | x | x | x | x |  |  |  |  | - | x | x | x | x | x | x | x |  |  |  |
| MOSES – TRIFFID | T | x | x | x | x | x |  |  |  |  | - | x | x | x | x | x | x | x |  |  |  |
| ORCHIDEE | T | x | x | x | x | x | x |  |  |  | - | x | x | x | x | x | x | x |  |  |  |
| SEIB-DVGM | T | x | x | x | x | x |  |  |  |  | - | x | x | x | x | x | x | x |  |  |  |
| VISIT-e | T | x | x | x | x | x |  | x |  |  | - | x | x | x | x | x | x | x | x | x |  |
| JSBACH3.2 | T | x | x | x | x | x |  | x |  |  | - | x | x | x | x | x | x | x |  |  |  |
| HAT | T | x | x | x | x | x |  |  |  |  | - | x | x | x | x | x | x | x |  |  |  |
| CLM4 | T | x | x | x | x | x |  | x |  |  | - | x | x | x | x | x | x | x |  | x |  |
| JULES | T | x | x | x | x | x |  | x |  |  | - | x | x | x | x | x | x | x | x |  | x |
| WOMBAT | M |  |  |  | x | x | x | x |  | x |  | - |  |  |  | - |  | - |  |  |  |
| iBGC | M |  |  |  | x | x |  |  |  | x |  | - |  |  |  | - |  | - |  |  |  |
| CMOC | M | x |  |  | x | x | x | x |  |  |  | - |  |  |  | - |  | - |  |  |  |
| BEC | M |  |  |  | x | x | x | x | x | x | x | - |  | x |  | - |  | - |  |  |  |
| MARBL | M |  |  |  | x | x | x | x | x | x | x | - |  | x |  | - |  | - |  |  |  |
| BFMv5.1 | M |  | x |  | x | x | x | x | x | x | x | - |  | x |  | - | x | - |  |  |  |
| PISCESv2 | M |  |  |  | x | x | x | x | x | x | x | - |  | x |  | - |  | - |  |  |  |
| PISCESv2-gas | M |  |  |  | x | x | x | x | x | x | x | - |  | x |  | - |  | - |  | x | x |
| TOPAZ2m | M |  |  |  | x | x | x | x | x | x | x | - |  | x |  | - |  | - |  |  |  |
| Diat-HadOCC | M |  |  |  | x | x | x | x |  | x | x | - |  |  |  | - |  | - |  |  | x |
| NPZD | M |  |  |  | x | x | x | x |  |  |  | - |  |  |  | - |  | - |  |  |  |
| OECO-v2 | M |  |  |  | x | x | x | x | x | x |  | - |  |  |  | - |  | - |  | x | x |
| HAMOCC6 | M |  |  |  | x | x | x | x | x | x | x | - |  |  |  | - |  | - |  | x | x |
| unnamed | M |  |  |  | x | x | x | x | x |  |  | - |  |  |  | - |  | - |  |  |  |
| HAMOCC5 | M |  |  |  | x | x | x | x | x | x | x | - |  |  |  | - |  | - |  | x | x |
| MEDUSA | M |  |  |  | x | x | x | x |  | x | x | - |  |  |  | - |  | - |  |  | x |