# Peer review of "ESD Reviews: Evidence of multiple inconsistencies between representations of terrestrial and marine ecosystems in Earth System Models"

_Earth System Dynamics, 2020_

## Referee Comment (RC1) · Anonymous Referee #1 · 27 Sep 2020

Comments on "Evidence of multiple inconsistencies between representations of terrestrial and marine ecosystems in Earth System Models" by F. Pellerin et al.

This manuscript reviewed how biogeochemical, biogeophysical, and particle exchange processes of marine and terrestrial ecosystems are implemented in the contemporary Earth System Models. The authors focused on climatic feedbacks of these biospheric processes and clarified which processes have inconsistencies between the marine and terrestrial ecosystems.

[Figure]

General points As revealed by model-intercomparison studies, the contemporary Earth System Models have serious uncertainties in the representation of climatic feedback mechanisms and thus future climate projections. The present manuscript addressed, in this regard, an important issue to reduce the uncertainties, making contributions for our climate management. However, this is a narrative review and lacks quantitative analysis. Therefore, in my view, this is not for expert researchers of the study area but for students or researchers of other expertise.

The authors tried to contrast between marine and terrestrial ecosystems and to specify inconsistencies between the ecosystems. It is not obvious for me that the two systems should be represented in a consistent manner, because they have clear differences in physical and chemical properties. For example, lateral and vertical convective transportations are important for marine ecosystems, while these transportations exert relatively small roles in terrestrial ecosystems. The authors should, at first, clarify the similarity and difference between the marine and terrestrial ecosystems. Apparently, these issues have been addressed by ecological and meteorological studies, and then a brief summary is sufficient. Another caveat on this manuscript is the lack of consideration on the interaction between marine and terrestrial ecosystems, such as riverine transportation and coastal system, which become increasingly important in the present Earth System Model studies.

Finally, I conclude that the manuscript needs major revision before being accepted for publication. Introduction should provide more research background, and Discussion should provide more insightful discussion.

Specific points

Line 91: "THOM" should be "Thom".

Line 102: The gas and particle section has some overlap with the biogeochemical section; this section should, for example, focus on short-lived species such as BVOCs and organic aerosols. Similarly, in Figure 2, "respiration" appears in categories 1 and

3.

Line 230: Recently, a synthesis on the global CH4 budget (Saunois et al., 2020) was published.

Line 261: This Discussion and Conclusions section should provide more in-depth discussions such as priority for improvement of Earth System Models.

Line 292: A new paper on TRY (Kattge et al., 2020) was published.

References

Kattge, J., et al.: TRY plant trait database - enhanced coverage and open access, Global Change Biol., 26, 119–188, 10.1111/gcb.14904, 2020.

Saunois, M., et al.: The global methane budget 2000–2017, Earth System Science Data, 12, 1561–1623, 10.5194/essd-12-1561-2020, 2020.

---

## Referee Comment (RC2) · Anonymous Referee #2 · 22 Oct 2020

General comments:

The manuscript starts from the premise that representations of terrestrial and marine ecosystems differ in present ESMs and that this raises the question whether inconsistencies between terrestrial and marine ecosystem models exist, with consequences on the simulated effects on climate.

I find it difficult to follow this premise. Terrestrial and marine ecosystems ARE different. Competition for space and water is much different, the amount of structural plant

biomass is different, the role of physico-chemical effects like gravity or fires is different, and trophic interactions are different. Why should descriptions of such different systems be similar in models?

The authors assume that 'Inconsistencies in ecosystem representation can lead to wrong predictions about the role of the biosphere in the climate system' (l.270). This appears to be a main motivation for their study, but there is no evidence provided that this statement may be correct. Also, there is no justification for their conclusion that a unified framework of terrestrial and marine ecosystem models is urgently needed (l.288) or that combined databases of terrestrial and marine organisms (l.289) would be useful.

I am not at all convinced that there would be any improvement in trying to model the different terrestrial and marine systems via a unified model framework. By introducing processes of secondary importance, noise level, parameter indeterminacy and thus model uncertainty may grow. Models are simplifications of the real world and should not be made more complex than necessary. The authors provide no evidence that their suggestions (which they call conclusions) can be useful. I don't think that this is a useful scientific approach.

specific points:

l.78: The statement that the biological (carbon) pump is responsible for 90% of the marine carbon uptake is wrong (and never said by Friedlingstein et al., 2019). Current marine carbon uptake is essentially exclusively due to the solubility pump. I am not aware of any solid evidence that the biological carbon pump is playing a significant role in the oceanic uptake carbon in the anthropocene.

l.169: photosynthesis and respiration are not ignored in marine ecosystem models.

They are rather treated as a constant proportion of NPP. This difference wrt the representation of terrestrial plants may well be justified by the tighter stoichiometry in marine plant biomass due to the absence of major investment into cellulose.

Fig.1: The term 'Redfield ratio' is used incorrectly. Presumably 'C:N:P ratio' is meant here (which can vary - the Redfield ratio is a fixed C:N:P ratio that does not vary).

––––––––––––––––––––––––––––––

---

## Author Comment (AC1) · 27 Oct 2020

First of all, we would like to thank the reviewers for their time to read the manuscript and to send us thought-provoking comments.

Both reviewers question our approach for a common framework to describe marine and terrestrial ecosystems in a consistent way. They argue that both ecosystems are different and thus there is no value in treating them in an integrative way. Of course, we agree that there are important differences between terrestrial and marine systems, and

we will make this more clear in the revised version of the manuscript. However, many biological processes occurring in these systems are identical, and these processes have an important effect on climate through their role in biogeochemical cycles and biogeophysics. Why should we represent identical biological processes in a different way? We argue that there is no convincing reason why the same processes should be treated differently for marine and terrestrial systems.

We do believe that the unjustified difference in the representation of important biological processes between ecosystem models is an issue. For instance, photosynthesis and respiration do have a different thermal reaction norm. While this is taken into account in terrestrial, marine ecosystem models ignore this fact as illustrated in numerous experiments (see e.g. Wohlers et al. 2009, Padfield et al. 2016, Schaum et al. 2017). Thus any warming scenario will at its best simulate the consequences for carbon storage correctly for terrestrial ecosystems and inaccurately for marine systems. In coupled model runs this error will propagate in the system but we have currently no idea about the size of the error. Only with a unified model framework we will be able to address this question. Similarly, terrestrial models represent various biogeophysical mechanisms (i.e. albedo, light absorption, roughness length), while marine ecosystem models poorly represent them. In the case of light absorption, it has been demonstrated by coupled ocean ecosystem models that this feedback has major impact on climate predictions (see Patara et al. 2012 for a review). Again, in ESM runs, this error will propagate and might impact model predictions, in particular regarding the relative role of terrestrial vs marine ecosystems in the climate system.

In conclusion, the present study assesses the current inconsistencies in the representation of the same processes and mechanisms between marine and terrestrial ecosystem models. Such an assessment is a prerequisite to generate a common framework and a more quantitative analysis on the current inconsistencies in marine and terrestrial ecosystem models in ESMs.

Please note that all the other specific points identified by the reviewers are not discussed here but will be addressed in the revised version of the current manuscript.

References:

Wohlers, Julia, Anja Engel, Eckart Zöllner, Petra Breithaupt, Klaus Jürgens, Hans Georg Hoppe, Ulrich Sommer, and Ulf Riebesell. 2009. "Changes in Biogenic Carbon Flow in Response to Sea Surface Warming." Proceedings of the National Academy of Sciences of the United States of America 106 (17): 7067–72. https://doi.org/10.1073/pnas.0812743106.

Padfield, Daniel, Genevieve Yvon-Durocher, Angus Buckling, Simon Jennings, and Gabriel Yvon-Durocher. 2016. "Rapid Evolution of Metabolic Traits Explains Thermal Adaptation in Phytoplankton." Ecology Letters 19 (2): 133–42. https://doi.org/10.1111/ele.12545.

Schaum, C. Elisa, Samuel Barton, Elvire Bestion, Angus Buckling, Bernardo Garcia-Carreras, Paula Lopez, Chris Lowe, et al. 2017. "Adaptation of Phytoplankton to a Decade of Experimental Warming Linked to Increased Photosynthesis." Nature Ecology and Evolution 1 (4): 1–7. https://doi.org/10.1038/s41559-017-0094.

Patara, Lavinia, Marcello Vichi, Simona Masina, Pier Giuseppe Fogli, and Elisa Manzini. 2012. "Global Response to Solar Radiation Absorbed by Phytoplankton in a Coupled Climate Model." Climate Dynamics 39 (7–8): 1951–68. https://doi.org/10.1007/s00382-012-1300-9.